# A Cluster Analysis of University Commuters: Attitudes, Personal Norms and Constraints, and Travel Satisfaction

**DOI:** 10.3390/ijerph18094592

**Published:** 2021-04-26

**Authors:** Marco De Angelis, Luca Mantecchini, Luca Pietrantoni

**Affiliations:** 1Department of Psychology, University of Bologna, Via Berti Pichat 5, 40126 Bologna, Italy; luca.pietrantoni@unibo.it; 2Department of Civil, Chemical, Environmental, and Materials Engineering, University of Bologna, Viale del Risorgimento 2, 40136 Bologna, Italy; luca.mantecchini@unibo.it

**Keywords:** cluster analysis, sustainable mobility, travel satisfaction, commuting, psychosocial factors

## Abstract

Higher education institutions are recognised as settings where the community’s awareness of sustainable mobility can be strengthened or reshaped. The first objective of the present study was to identify groups of commuters based on their modal choice in a large higher education institution in Italy. The second objective was to compare the groups on socio-demographic and psychosocial variables, specifically attitudes, personal norms, personal constraints, and travel satisfaction. The cluster analysis revealed five different types of commuters: car-oriented, two-wheeled urban users, pedestrians, long-distance commuters, and regular bus users. Attitudes, personal constraints and norms, and satisfaction differed in the five groups of commuters. The present study provides insights for behavioural change programmes and organizational policies on sustainable mobility.

## 1. Introduction

National and local authorities are increasingly developing and applying measures and strategies to reduce the environmental impacts of daily commuting, promoting active and sustainable transport modes. The possibility of designing sustainable mobility models together with organisational practices is a matter of social and institutional responsibility or sustainability goals and personal and organisational well-being. There is a growing consensus that organisations should actively contribute with policies to promote sustainable behaviours [1].

A plethora of different psychosocial factors influences mode choice, such as values, attitudes, and personal norms and constraints. Psychosocial factors are helpful to obtain a better understanding of mode choice behaviour [2,3,4]. Indeed, mobility choices reflect users’ beliefs and perception, attitudes towards different means of transport, habits, and subjective and objective constraints. Therefore, scientific literature refers to travel behaviour [5], a definition encompassing both knowledge on traffic, built environment characteristics, and social and psychological aspects concerning decision theory. For example, attitude has been identified as the most significant factor influencing the intention to use public transit [6,7]. Travel characteristics such as travel distance or time also influence mode choice [8]: the longer the distance to be covered, the higher the tendency to use the car. In terms of social variable, previous research supports a recent trend concerning the younger generation [9], where the Western world’s students are more willing to use other modes of transport than the car for their trips.

From a psychological perspective, mobility behaviour patterns are shaped by relatively stable intentions, norms, or attitudes [10,11]. The theory of planned behaviour (TPB) [12] and the value-belief-norm theory (VBN) [13] are among the most influential and well-documented models in the literature. The TPB postulates that particular behaviour (i.e., use of alternative means of transport to the car) is significantly determined by the perceived behavioural control, that is, the individual perceived ability to perform the action. In other words, at a higher level of perceived behavioural control the individual is more prone to adopt a specific behaviour (e.g., commute by active mode choice). In addition, the VBN highlighted the critical role played by a personal feeling of moral obligation towards the environment (e.g., personal norms) that trigger and drive the behaviour (e.g., reduce the use of the private vehicle).

Several studies provided strong empirical support for both models in the field of transport mode choice [3,7]. A recent systematic review with meta-analysis [14] highlighted the most robust correlations of mobility choices on intentions, perceived behavioural control, and attitudes. For example, favourable attitudes towards alternative transport modes turned out to be negatively associated with private vehicles. In terms of attitudes, whereas flexibility has been seen as a predictor of car use, the priority of convenience significantly affected the intention to use public transport [15]. Furthermore, Abrahamse et al. [16] showed that car choice was mainly explained by perceived behavioural control and personal values defined the intention to reduce car use. However, when considering mode choice determinants and decision-making processes, situational circumstances or personal constraints might impact the adoption of sustainable means of transport. For example, car access [7], comfort [17], children [18], or vulnerable older adults to take care of [19] are considered personal constraints that might prevail in mode choice decision-making.

In repetitive and habitual behaviours such as car use or commuting trips, habits play a crucial role [2,20]. In other words, considering the daily mobility behaviour that tends to occur in a stable context, commuter mode choice may be considered a habitual response to a stable environment. From a behavioural change perspective, it can be argued that a repetitive and regular mode choice (e.g., commuting by car) might be challenging to switch, so the user reconsiders his or her mental model (e.g., commuting by alternative mode options). A series of studies by Verplanken and colleagues [21,22] into the habit discontinuity hypothesis have demonstrated how contextual changes, for example, moving homes or offices, can weaken habits and create “a window of opportunity” where people can redefine their mobility pattern and embrace a behavioural change. The event may involve a lasting change, such as relocating a company’s headquarters [21]. Brown et al. [23] found that a temporary lack of parking at the university led some motorists to opt for the light rail and, for some of them, this mobility behaviour became a long-term choice.

Although much has been written about these factors, few studies provided insights into how these complex and interconnected factors might affect mode choice and decision-making patterns.

Higher education institutions have become more conscious of their performance on sustainability. Universities and academic institutions are recognised as settings where the university community’s awareness on sustainable mobility can be strengthened or reshaped. University programmes aim at leveraging staff/student behavioural change and wider institutional change by starting with sustainable transport [24]. Furthermore, universities provide a unique context for behavioural research because they are friendly to alternative travel modes, have a higher density than other contexts, and offer mixed travel modes [25]. Mobility behaviours of university students and staff have been investigated by considering different variables. Personal, household, density, diversity, and design estimates are related to active travel (bicycling, walking, and mass-transit modes) among a university commuter population. Still, distance cannot be considered a universal barrier to active travel potential [19,20,21,22,23,24,25,26]. Other studies highlighted differences in travel behaviour between students living on campus and students living off campus [25] or explored the role of personal motivators and barriers as determinants for the active travel choices of university students [27]. A study by Szmelter-Jarosz and Suchanek [28] aimed to examine the travel behaviour of young Polish students. Factor analysis allowed the attitudes towards travelling among those young adults (Y Generation, Y’s, Y Gen) to be grouped. Three factors were identified, and they were associated with luxury and self-expression, freedom and comfort, safety, and environmental friendliness. In general, studies examining commuting patterns, barriers, and motivators affecting transport decisions in the university population show an increasing potential for active modes. Reducing actual and perceived travel time by bus and bicycle would greatly impact commuting patterns [29,30]. Politis [31] pointed out the factors contributing to the acceptance of sustainable mobility measures for university commuters by means of factor analysis and SEM, highlighting the priority role played by individual travel choices (car dependency). The work of Papaioannou and Politis [32] examined the impact produced by endogenous characteristics of commuters belonging to a large university community on their travel mode choices, combining the behavioural change model with the discrete choice modelling approach.

Although attention has been paid to travel satisfaction studies during the last decade, less research has investigated the different dimensions of commuting satisfaction [33,34]. Gerber et al. [35] examined the motivations and barriers involved in the evolution of commuting characteristics in a university centre. The change in travel time was an essential factor in commuter satisfaction, whereby reduced commuting time improved satisfaction, as expected. Travel-related attitudes influence satisfaction with travel more than socio-demographic attributes [34]. Public transport users are globally more satisfied with commuting than car drivers. The effect of attitudes and other latent constructs is paramount, even concealing most socio-demographic attributes to assess satisfaction. Furthermore, active modes (walking and cycling) induce higher satisfaction with travel time and lower dissonance, whereas public transit riders tend to be more satisfied but also more dissonant compared to automobile users [36,37,38,39].

Previous studies have grouped travellers under different behavioural, socio-economic, demographic, and psychosocial variables to investigate the underlying motivations, personality traits, attitudes, perceptions, travel experiences, and travel satisfaction. Anable [40] grouped transport users, distinguishing subgroups among car owners (i.e., malcontent motorists, complacent car, addicts, die-hard drivers, aspiring environmentalists) and non-car owners (i.e., carless crusaders, reluctant riders). The study recognised the need for tailored strategies when promoting a modal change, specific to each typology of users, with individual values, attitudes, and preferences.

A different methodological approach can be found in Diana and Mokhtarian [41]. The authors grouped transport users based on their travel behaviour through objective measure, such as their typical weekly mileage of different modes of transport: (1) driver or passenger in any personal vehicle, (2) bus, (3) rail, and (4) walking, jogging, or cycling. Comparisons between levels of car usage and public transport modes revealed that socio-economic characteristics alone could not explain travel patterns. Similarly, Molin and colleagues [42] clustered road users based on their self-reported frequency of mode use. The latent class cluster analysis identified (multi)modal travel groups (i.e., the car multimodal users; the multimodal bike group; the bike and car group; the mostly car group, and the public transport group), thus exploring the effects of socio-demographic and attitudinal dimensions towards each mode of transport. Recently, a cluster analysis was performed on cycling commuters [43], and three clusters were identified: informal workers with children, short-distance students, and occasional cyclists. The clusters were based on household composition, employment status, and frequency of cycling to work/school.

From our synthetic convergent review, we identified some research gaps. Previous studies developing a sustainable mobility model based on psychosocial factors in profiling travellers and classifying their mobile behaviour are limited. Specifically, attitudes, personal norms and constraints, and travel satisfaction can explain commuters’ behaviours and sustainable transport choices in university settings, but their role has not been elucidated. Therefore, the first objective of the present study was to identify groups of commuters based on their modal choice in a large higher education institution in Italy. The second objective was to compare the groups in terms of socio-demographic and psychosocial variables, precisely, attitudes, personal norms and constraints, and travel satisfaction.

## 2. Materials and Methods

### 2.1. Procedure

Ethical approval for the study was obtained from the Ethical Committee of the University of Bologna. The online survey was administered to the academic community through an email invitation (90,488 among students, professors, and staff). A link to the online questionnaire was also published on the University Sustainable Mobility website, where participants could access information about the purposes of the research, data protection, and privacy issue statements.

### 2.2. Participants

A total of 11,773 participants from the academic institution filled out the questionnaire. Only participants who completed the entire survey were included, leaving a sample of 8093 participants included in the analysis (actual response rate, 8.94%). Of these, 4798 (59.3%) were female. In the present study, the sample was primarily composed of students (76.3%) and young people who had left their hometown (40.8%) to undertake, continue, or conclude their university path. Professors represented 11.6% of the population, followed by administrative staff (7.5%) and personnel directly involved in different research activities under project contracts such as research grants or scholarships (i.e., fixed-term academic collaborators, 4.6%). The sample mean age was 27.01 years old (SD = 11.81).

### 2.3. Campus Location, Connections, Transport Infrastructures, and Services

The university has established five campuses in the main provinces of the Emilia-Romagna Region. These urban cities, where the academic population is an integral part of the population of that metropolitan area, are well connected by roadway and railway network and regular public transit services. The five campuses are connected by direct railway lines, both high-speed and regular trains, with an average scheduled frequency of 30 min. The time necessary to cover the distance between Bologna, the main seat of the university, and other campuses ranges from 56 min by high-speed train to 1 h 31 min by regular train. As far as public transit is concerned, the academic community can benefit from an extensive network of urban and inter-urban connections.

Moreover, the university offers a discounted public transport season to all students and academic workers. In terms of bicycle transportation, cities where campuses are located have extensive networks of bike lanes and various bike and e-bike sharing operators. Finally, road networks are extensive and provide easy access to the multiple campuses, which generally have reserved parking for private cars, except when located in limited traffic zones.

### 2.4. Measures

Participants completed an online questionnaire. The questionnaire included demographic measures such as age, gender, academic status (i.e., resident students, non-resident students, professors and fixed-term academic collaborators, administrative staff), and nationality. In addition, participants indicated whether they had a private car (Y/N) or a bicycle (Y/N) available for their trips to the destination and whether they had purchased a public transport pass (Y/N), regardless of the type of public transport (e.g., bus or train).

Participants were asked to provide a detailed description of their commuting trip. Respondents reported each step of their modal strategy by defining both the travel mode used and an estimation of the time and distance covered from home to the destination. We did not provide a cut-off length threshold to define when a travel mode should or should not be included in the participant’s travel chain (e.g., less than 200 m), which may have led to a lack of understanding regarding the walking mode. Nonetheless, as a result, it was possible to explore the most widely adopted travel chains and the proportion of use of each mode within the entire daily travel chain.

Participants were asked whether they experienced a relocation or residential change during the last two years. Other psychosocial measures were collected.

Personal constraints in sustainable mobility. The five-point Likert scale (from 1 = completely disagree to 5 = completely agree) adapted by Klöckner and Blöbaum [2] measures personal constraints in using sustainable modes of transportation. The scale was composed of three items. An example item was “It would be difficult to manage my trips with environmentally friendly means of transportation.” Cronbach’s alpha was 0.767.

Personal norms. Participants expressed their degree of agreement (from 1 = completely disagree to 5 = completely agree) regarding personal norms about sustainable mobility. The scale was composed of three items. An example item was “When I have to choose which means of transport to take, I always try to use a sustainable mode of transport.” Cronbach’s alpha was 0.792.

Attitudes towards different transport modes. Participants evaluated public transport/car/bicycle according to various characteristics using a five-point Likert scale (from 1 = not at all to 5 = very much). The three questions for each transport mode were “Considering the public transport/car/bicycle, how much do you consider this mode of transport as relaxing/reliable/fast?” Cronbach’s alpha for attitudes towards public transport, the car, and the bicycle were 0.866, 0.802, and 0.804, respectively.

Travel satisfaction. Five items on a 10-point Likert scale sought to assess participants’ overall satisfaction with their travel, personal satisfaction with the trip length, cost, perceived comfort, and perceived safety. An example of an item was “From 1 (not really) to 10 (fully), how satisfied are you with the following aspects of your home–university trip?” Cronbach’s alpha was 0.844.

### 2.5. Statistical Analysis

A two-step cluster analysis was performed using both categorical (i.e., number of modes of transport within the entire trip) and continuous (i.e., trip length and duration, the ratio of means of transport usage) variables related to the mode choice. Widely used in travel and transportation research, the added value of a two-step cluster analysis is the opportunity to automatically determine the number of clusters while properly handling categorical and continuous data simultaneously. Furthermore, this type of analysis is highly suggested when dealing with large datasets [44].

Based on findings and inputs from Molin et al. [41,45], the current study aimed to identify groups of travellers based on their commuting travel behaviour and their primary mode of transport instead of segmenting travellers based on their demographic, attitudinal, habitual, and normative factors. Therefore, the objective (i.e., duration, trip length, number of modes) and behavioural variables (i.e., the share of modes use) were adopted to cluster the academic commuters’ sample to highlight typical travel chains. As a first step, data are clustered using log-likelihood distance as a similarity criterion. In other words, the analysis progresses by sequentially establishing clusters, either pre-existing or entirely new, with an increasingly higher log-likelihood value. The second step of the analysis consists of combining the previously created clusters according to an agglomerative hierarchical logic. Finally, the goodness-of-fit analysis conducted investigating the silhouette measure of cohesion and separation ensures that clusters are distinctly heterogeneous among themselves and as homogeneous as possible within them. Values of 0.50 or more reflect a good fit [46].

Chi-square tests and ANOVAs were performed on additional categorical and continuous variables (i.e., socio-demographic, attitudinal, psychosocial) to confirm the significance of the differences between the segments created. The Bonferroni adjustment was adopted for further interpretation of the cluster solution. The interpretation of the results was based on both statistical significances (*p* < 0.05) and measures of effect size based on Cramer’s V and η^2^ [47]. The data were analysed using IBM SPSS 26.

## 3. Results

### 3.1. Cluster Analysis

The two-step cluster analysis yielded a five-group solution (Table 1). The average Silhouette coefficient was 0.80, indicating excellent cohesion and separation. The ratio between the most extensive and smallest cluster was 2.36, indicating balanced cluster sizes.

The first group (*n* = 2305, 28.5% of the respondents) was characterised by commuters who mostly relied on the train (average of 84% of the entire trip) to reach their destination. Respondents in this group tended to adopt a multimodal strategy; 73.6% of participants changed the mode of transport three times along the commuting trip. The other stages of the route were covered by car (7%), by bus (2%), or on foot (1%). People in this group needed to travel long distances to reach the destination (M = 63.27 km; SD = 31.69). This group was labelled “long-distance commuters.”

The second group (*n* = 2211, 27.3% of the respondents) was characterised by being multimodal with a clear preference for the bus (83%), whereas the rest of the trip was usually covered on foot (14%). People in this group usually travelled about 10 km to reach their destination (M = 9.85 km; SD = 10.78), taking approximately one hour for the round commuting trip. This group was labelled “regular bus users.”

The third group (*n* = 974, 12.0% of the respondents) was characterised by commuters who preferred a private car for commuting. Commuters in this group usually travelled about 25 km to reach the destination (M = 24.75 km; SD = 27.27), taking approximately one hour for the round-trip commute. This segment was labelled “car-oriented commuters.”

The fourth group (*n* = 1091, 13.5% of respondents) was characterised by people traveling a short distance (M = 3.61 km; SD = 4.69) and using mostly cycling (85%) or travelling by motorcycle (14%). The destination was typically 3.61 km distant from home. The total duration of the trip was about half an hour. This segment was labelled “two-wheeled urban users.”

The final group (*n* = 1512, 18.7% of the respondents) was characterised by people traveling a short distance (M = 1.69 km; SD = 2.30) and walking to reach their destination. The destination was 1.68 km from home, taking approximately half an hour for the round trip. This segment was classified as “pedestrians” since walking represented the only mode.

All comparisons between groups of segmentation variables were significant at *p* < 0.001, with large effect sizes.

### 3.2. Comparisons among the Five Commuter Groups

Table 2 shows the differences between groups of travellers in terms of gender, age, academic role, availability of transport modes, and relocation. Commuters grouped as car-oriented and two-wheeled urban users tended to include more males than the other traveller groups. In terms of age, commuters classified as car-oriented were older whereas pedestrians were younger. Resident students showed a higher propensity to use public or private modes of transport. Professors and administrative staff expressed a preference for the use of private transport, whereas non-resident students predominantly adopted active or two-wheeled modes of transport.

The five groups showed different percentages considering the availability of a car, a bicycle, and easy access to public transport. Less than 50% of members included in the regular bus users, two-wheeled urban users, and pedestrian groups had access to a car. Pedestrians represented the group with the lowest percentages of accessibility to both cars and bicycles. Long-distance commuters and regular bus users consisted almost entirely of people with a public transport pass. Regular bus users, two-wheeled urban users, and pedestrians reported more experiences of relocation in the last two years. All comparisons between the groups were significant at *p* < 0.001, with moderate-to-high effect sizes for all variables except gender.

Table 3 shows comparisons of attitudes, personal constraints and norms, and satisfaction in the five groups of commuters. Figure 1 provides a comparison of the standardised scores for kilometres travelled, journey time, and overall satisfaction among the five groups of travellers. In Figure 2, differences in satisfaction with accessibility, comfort, cost, infrastructure, journey duration, and safety are shown through standardised scores.

Participants showed a favourable attitude towards the car, regardless of the group they belonged to. Attitudes towards cycling and public transport were moderately positive. Two-wheeled urban users and pedestrians were more likely to show positive attitudes (fast, relaxing, and flexible to personal needs) towards the bicycle compared to other groups. Attitudes towards public transport were moderately negative in all groups, in particular in car-oriented, long-distance commuters, and two-wheeled urban users. Car-oriented commuters reported low scores in attitudes towards any means of transport other than the car.

Regular bus users, two-wheeled urban users, and pedestrians exhibited fewer personal constraints to sustainable mobility. Two-wheeled urban users and pedestrians scored significantly higher in personal norms than other groups of travellers. In terms of personal satisfaction for the overall commuting experience as well as for specific aspects of the trip, pedestrians and two-wheeled urban users were more satisfied with their route compared to all the other groups. Nonetheless, the latter reported a significantly lower level of comfort and perceived safety than the other groups of travellers. Long-distance commuters expressed a significantly lower level of satisfaction about their route than all other segments, in particular for the duration of their trip, the related costs, and the overall perceived comfort. All comparisons between the groups were significant at *p* < 0.001, with moderate-to-high effect sizes for all variables except attitudes towards the car.

## 4. Discussion

In the present study, the first objective was to identify groups of commuters based on their modal choice in a large higher education institution in Italy. The second objective was to compare the groups in terms of socio-demographic and psychosocial variables. This elucidated commuters’ decision-making processes and the role of attitudes, personal norms, and constraints in the adoption of sustainable mobility.

The cluster analysis revealed five different types of university commuters. The first type was represented by “long-distance commuters.” They were characterised by commuters who tended to adopt a multimodal strategy, relying mostly on the train to reach their destination. This group of commuters was typically represented by resident students who lived with their family or partner, had access to the car, and tended to own a public transport pass even though their attitudes towards this mode of transport were among the most negative compared to the other group of travellers. Indeed, flexibility, comfort, and reliability were some of the most critical aspects highlighted by commuters included in this category. Commuters who used the train to cover long distances were the most dissatisfied. De Vos [33] depicted these users as “dissonant users,” where about half of the participants who claimed to use public transport expressed an evident aversion towards this travel mode. In other words, half of the respondents were not travelling with the preferred travel mode. In the present study, the multimodal travel strategy adopted (>70% trimodal) might itself explain the observed dissatisfaction in terms of lack of quality of modal transfers at interchange nodes [34]. In other words, the perception of the overall quality of multimodal travel might be lower than observed for users belonging to other distance classes and adopting other modal strategies due to the interchange nodes.

The second type was represented by “regular users.” They tended to use the bus and to walk, with the latter probably adopted to get to the bus stop or the destination once off the bus. They were mostly youngsters, and they tended to have negative attitudes towards public transport. In general, dissonant travellers seemed to confirm what has recently been debated about the relationship between attitudes and behaviour. The idea of a positive relationship between attitudes and modal behaviour seems to be partly disconfirmed among public transport users [42]. This discrepancy can be explained by factors such as the perception of control of one’s behaviour or specific characteristics of the built environment. For example, being too far from the destination could force people to adopt transport modes that they do not prefer. In the present study, accordingly, people who lived far from the destination tended to adopt public transport even if it was perceived negatively. Comfort, cost, and duration seemed to be key attributes that did not meet the user mobility needs of the home–work route. The perceived value of the public transport choice, if neglected, can only discourage the use of this mode, further aggravating the environmental context. In other words, if the use of public transport is forced by long distances to be covered or by the absence of a private vehicle, it will lay the basis for a dissatisfied user who will change mode as soon as one of the conditions is resolved (e.g., buying a car). The role played by the instrumental attributes (e.g., comfort, accessibility, reliability) of public transport in attracting car users is highlighted here. Further strategies relate to the possibility of providing internet services to use available time for other activities such as work, study, or leisure activities or to obtain more real-time information about the situation of public services by reducing scheduling costs.

The third type was represented by “private car-oriented commuters.” They were older and predominantly males. They showed positive attitudes towards the use of private vehicles. Personal overall satisfaction about their commuting trip was moderate and, apart from the comfort experienced by using the car, ratings of other aspects, such as trip duration, safety, and cost, were neither significantly negative nor positive. In the present study, age seemed to play a significant role in shaping the decision-making process of commuters. As people grow old, they tend to rely on driving. Some changes may occur throughout a person’s life that lead to greater reliance on a private vehicle as they get older [48,49]. The present results can help in understanding the tendency of the new generation to prefer the use of greener modes. One of the most discussed points concerns the possibility that this trend is only delayed because today’s young people may have less chance of buying a car than previous generations [50]. Based on our findings, the younger generation might not use a car because they do not have access to it or do not have the possibility to use it because they have moved away from their place of residence to study. In other words, the decision to adopt the car as their primary mode, nowadays, is only postponed. Other authors support the possibility of a new generation more inclined to express inherently different attitudes and pro-environmental values, which manifests in more sustainable-friendly lifestyle choices over time [48,49]. In the present study, both positions seemed to coexist with young students (mainly female), on the one hand, showing greener values and a higher likelihood to use active modes of transport, and on the other hand, being more likely to use the resources available to them (i.e., access to a car). Nonetheless, as people grow older and possibly have greater economic availability from being actively working, getting married, or having children (e.g., professors, administrative staff), they tend to rely more on this mode of transport. In this sense, the university ought to take the responsibility to promote pro-environmental values as much as possible by emphasising reinforcement strategies, especially towards young people. Future studies on this topic are strongly encouraged, as preventing the development of a pro-car habit among young people would only risk exacerbating the current problem.

The fourth group (“two-wheeled urban users”) was characterised by people who usually covered short-medium distances (less than 10 km) to reach their destination. People in this group preferred to cycle or take their motorcycle/scooter. In this group, both male and females were equally represented, and non-resident students represented the majority of the commuters in this group. Previous studies found that bicycle use tends to be predominant among men. Prati et al. [51] conducted a survey with regular cyclists in six European countries, and gender differences in attitudes towards cycling tended to disappear. Indeed, within short and medium distances, cycling habits could be crucial to reducing car use. A lower level of satisfaction, on the other hand, was expressed in terms of perceived safety, which remains a barrier especially in mixed traffic environments. Indeed, actions and interventions have been focused on improving and expanding cycling infrastructure where policymakers can intervene and innovate. However, in urban areas, improvements in existing road infrastructure might be difficult due to the proximity of settlements to the network. An alternative way to increase the network level of service (i.e., safety) is to improve the comfort experiences on the road, making better use of the existing roads [52]. Indeed, in the present study, regular bicycle commuters expressed a high level of satisfaction in terms of comfort, cost, and durability, all instrumental attributes on which promotional campaigns and strategies to encourage the adoption of this active mode of transport could be based. In addition, future studies can deepen our understanding of which factors compensate for cycling risk perception to such an extent that the use of this mode is worthwhile.

Finally, “pedestrians” were commuters covering short distances by walking. This group was mainly composed of students who had moved away from home on purpose and showed the lowest average age. They tended to have experienced a relocation in the last two years. This was in line with the habit discontinuity hypothesis [21]. It posits that contextual changes may disrupt people’s habits. A “window of opportunity” might open, giving people the chance to redefine their behaviours by considering other mobility patterns deliberately and rationally. Commuters in this group had no access to a private vehicle or bicycle because of the residential change. Even though their attitudes towards bicycles were positive, they found walking more effective. Short distance and low income (no access to a car or a bike) may be the main factors influencing this modal choice, as highlighted by previous work. Beyond situational constraints, this target group seemed to be distinguished by the relocation experienced. In this sense, the university assumes a central role in welcoming the mobility needs of these users, promoting reinforcement and rewarding strategies that favour the formation of a new mode habit.

The present study has some limitations that need to be acknowledged. First, the study concerns the exclusive use of self-report measures, which may be subject to reporting bias, even though other previous research on this topic also used an online survey. The cross-sectional design precludes causality or chronological order of changes. However, the present study contributes to the analysis of the psychosocial factors that can differentiate groups of university commuters.

The understanding of personal constraints and norms, attitudes, and satisfaction could be a winning strategy in fostering the adoption of more sustainable modes of transport. Car dependence can be effectively reduced, increasing the perception of safety, convenience, and feasibility of walking, cycling, and public transport [53]. Having a strong car-commuting habit decreases the probability of mode shift to a new sustainable mode alternative. For example, stimulating e-cycling may be most effective if targeted at specific groups who are not currently engaging in active travel and who are dissatisfied with car use [54].

However, it is noteworthy that bidirectionality exists in attitude behaviour relations in transportation. For examples, evidence of a feedback loop among car pride, car ownership, and car use has been found. Enabled with this ownership, individuals use the car more frequently for commuting and individuals who use their car more frequently have greater pride in their vehicle [55].

## 5. Conclusions

These findings advocate the prominence of relevant psychosocial mechanism of university commuters and extend the literature on personal norms and travel satisfaction. Our findings increase the theoretical knowledge about commuting modal choice determinants and may also provide insights for behavioural change programmes and organisational policies. In fact, organisational sustainability policies can affect environmental and sustainable attitudes and behaviours [56], promoting well-being and increasing productive behaviours [57].

University campuses are a microcosmos of the urban landscape and an excellent testbed for implementing and evaluating novel mobility policies [58]. They can become a prominent player in promoting sustainable mobility, providing the contextual and instrumental basis for the adoption of more sustainable travel behaviour. Based on the findings from the present study, strategies for promoting behavioural change that the institution can adopt may relate to the use of incentives or rewards for those who have shown to adopt sustainable trip chains. In addition, loyalty programmes or information kits on available discounts or sustainable routes, especially for recently relocated travellers, could be a successful strategy in creating new sustainable habits. Another strategy could be to encourage the academic community to plan events or interventions to make the workplace greener. Finally, considering car use and its determinants, the possibility of supporting families in their daily commute is an interesting field of research in which to explore possible modes of intervention. Organisational policies along these lines could incentivise the use of sustainable modes by users as well as increase their individual and organisational well-being.

The study offers suggestions to satisfy the needs of students and university staff members considering the perceived social norms and subjective satisfaction. The results obtained in this study may be of relevance to many stakeholders, particularly university mobility managers. They are likely to play an active role in the process of planning and management of mobility transformations to maximise the environmental and social benefits arising from the effective and targeted implementation of different travel mode alternatives.

## Figures and Tables

**Figure 1 ijerph-18-04592-f001:**
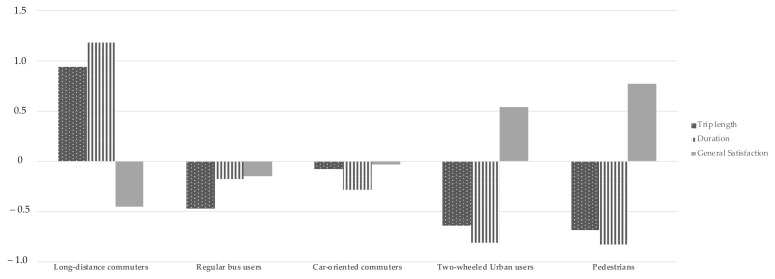
Standardised scores on trip length, duration, and general satisfaction among the five groups of travellers.

**Figure 2 ijerph-18-04592-f002:**
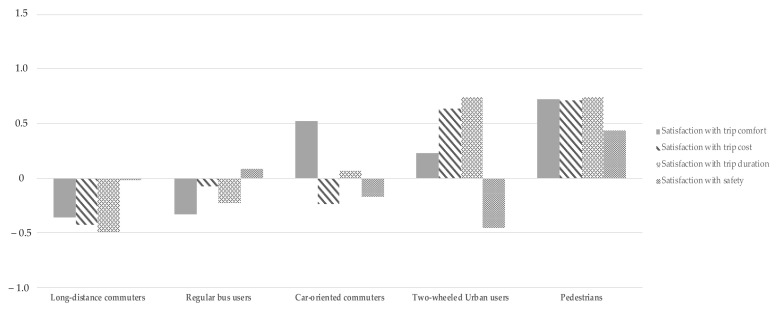
Standardised scores on satisfaction with accessibility, infrastructure, trip comfort, trip duration, and trip safety among the five groups of travellers.

**Table 1 ijerph-18-04592-t001:** Groups of travellers by the segmentation variables.

	Long-Distance Commuters (28.5%)	Regular Bus Users (27.3%)	Car-Oriented Commuters (12.0%)	Two-Wheeled Urban Users (13.5%)	Pedestrians (18.7%)	Segment Differences
Modal Strategy						*χ^2^*_16_ = 8284.20 *; *V* = 0.50
Unimodal	3.17 ^a^	17.23 ^b^	43.43 ^c^	98.63 ^d^	100 ^e^	
Bimodal	4.34 ^a^	35.14 ^b^	56.57 ^c^	1.28 ^d^	0 ^e^	
Trimodal	73.62 ^a^	44.28 ^b^	0 ^c^	0.09 ^c^	0 ^c^	
More than three modalities	18.87 ^a^	3.31 ^b^	0 ^c^	0 ^c^	0 ^c^	
Trip length (km) ^1^	63.27 ^a^	9.84 ^b^	24.74 ^c^	3.61 ^d^	1.68 ^d^	*F*_4,2586_ = 2586.01 *; *η* = 0.61
Trip duration (h) ^2^	2.68 ^a^	1.24 ^b^	1.12 ^c^	0.57 ^d^	0.55 ^d^	*F*_4,3543_ = 3380.39 *; *η*^2^ = 0.63
Average Ratio of Transport Mode Usage						
On foot	0.01 ^a^	0.14 ^b^	0.03 ^c^	0 ^a^	1.00 ^d^	*F*_4,8088_ = 37920.09 *; *η*^2^ = 0.95
Bicycle	0 ^a^	0 ^a^	0 ^a^	0.85 ^b^	0 ^a^	*F*_4,8088_= 9667.24 *; *η*^2^ = 0.83
Moto/Scooter	0 ^a^	0 ^a^	0 ^a^	0.14 ^b^	0 ^a^	*F*_4,8088_ = 288.43 *; *η*^2^ = 0.12
Car	0.07 ^a^	0.01 ^b^	0.96 ^c^	0 ^d^	0 ^d^	*F*_4,8088_ = 25418.93 *; *η*^2^ = 0.93
Bus	0.02 ^a^	0.83 ^b^	0 ^c^	0 ^c^	0 ^c^	*F*_4,8088_ = 26765.66 *; *η*^2^ = 0.93
Train	0.84 ^a^	0 ^b^	0 ^b^	0 ^b^	0 ^b^	*F*_4,8088_ = 48592.69 *; *η*^2^ = 0.96

*Note*. Values with different superscript letters (a,b,c,d,e) in the same row are significantly different from each other at *p* < 0.05. * *p* ≤ 0.001.^1^ Trip length refers to the one-way distance (in km) to reach the destination. ^2^ Trip duration (in hours) refers to both outward and return journey.

**Table 2 ijerph-18-04592-t002:** Gender, age, academic role, availability of transport modes, and relocation in the five commuter groups.

	Long-Distance (28.5%)	Regular Bus Users (27.3%)	Car-Oriented (12.0%)	Two-Wheeled Urban Users (13.5%)	Pedestrians (18.7%)	Segment Differences
Gender (%)						*χ^2^*_4_ = 94.282 *; *V* = 0.11
Female	62.4 ^a^	64.3 ^a^	56.0 ^b^	49.6 ^c^	66.0 ^a^	
Age, M ± SD	26.01 ^a^ ± 11.09	26.62 ^a^ ± 12.62	33.62 ^b^ ± 14.39	27.91 ^c^ ± 11.83	23.62 ^d^ ± 8.46	*F*_4,2849.98_ = 90.854 *; *η*^2^ = 0.05
Academic Role						*χ^2^*_16_*= 2577.50 *; V* = 0.28
Resident students	62.8 ^a^	31.5 ^b^	42.3 ^c^	18.5 ^d^	7.7 ^e^	
Non-resident students	16.6 ^a^	46.1 ^b^	10.7 ^c^	52.6 ^d^	80.7 ^e^	
Professors	7.7 ^a^	3.7 ^b^	18.5 ^c^	9.9 ^a^	3.9 ^b^	
Administrative staff	9.0 ^a^	14.6 ^b^	22.4 ^c^	12.5 ^b^	3.8 ^d^	
Research collaborators	3.9 ^a^	4.1 ^a^	6.2 ^a,b^	6.5 ^b^	4.0 ^a^	
Availability and Access						
Car	84.6 ^a^	43.3 ^b^	96.2 ^c^	45.4 ^b^	21.3 ^d^	*χ^2^*_4_ = 2302.98 *; *V* = 0.54
Bike	80.2 ^a^	55.0 ^b^	77.6 ^a^	94.3 ^c^	42.1 ^d^	*χ^2^*_4_ = 1140.62 *; *V* = 0.38
Public transport pass	93.0 ^a^	88.1 ^b^	18.1 ^c^	19.0 ^c^	20.4 ^c^	*χ^2^*_4_ = 4140.13 *; *V* = 0.72
*Habit Discontinuity (%)*						
Relocation or residential change during the last 2 years	19.8 ^a^	44.9 ^b^	18.4 ^a^	50.5 ^c^	67.8 ^d^	*χ^2^*_4_ = 1144.66 *; *V* = 0.38

*Note*. Values with different superscript letters (a,b,c,d,e) in the same row are significantly different from each other at *p* < 0.05. * *p* ≤ 0.001.

**Table 3 ijerph-18-04592-t003:** Comparisons of attitudes, personal constraints and norms, and satisfaction in the five groups of commuters.

	Long-Distance Commuters (28.5%)	Regular Bus Users (27.3%)	Car-Oriented (12.0%)	Two-Wheeled Urban Users (13.5%)	Pedestrians (18.7%)	Segment Differences
Attitudes Towards Different Modes of Transport						
Attitudes to car	3.56 ^a^	3.53 ^a,d^	3.79 ^b^	3.39 ^c^	3.47 ^d^	*F*_4,3352_ = 58.66 *; *η*^2^ = 0.02
Attitudes to bicycle	2.86 ^a^	2.82 ^a^	2.68 ^b^	3.39 ^c^	2.93 ^d^	*F*_4,3343_= 162.93 *; *η*^2^ = 0.07
Attitudes to public transport	2.34 ^a^	2.56 ^b^	2.20 ^c^	2.45 ^d^	2.52 ^e^	*F*_4,8032 =_ 93.00 *; *η*^2^ = 0.04
Personal Constraints and Norms						
Personal constraints	2.80 ^a^	3.59 ^b^	2.10 ^c^	3.91 ^d^	4.00 ^d^	*F*_4,3432_ = 1048.81 *; *η*^2^ = 0.31
Personal norms	3.52 ^a^	3.61 ^b^	3.00 ^c^	3.84 ^d^	3.79 ^d^	*F*_4,3316_= 116.00*; *η*^2^ = 0.05
Travel Satisfaction						
General	5.16 ^a^	5.93 ^b^	6.22 ^c^	7.68 ^d^	8.82 ^e^	*F*_4,3349_ = 640.95 *; *η*^2^ = 0.22
Duration	4.09 ^a^	4.89 ^b^	5.77 ^c^	7.78 ^d^	7.80 ^d^	*F*_4,3334_= 787.96 *; *η*^2^ = 0.26
Cost	4.97 ^a^	6.05 ^b^	5.55 ^c^	8.21 ^d^	8.45 ^d^	*F*_4,3263_ = 534.85 *; *η*^2^ = 0.21
Comfort	4.33 ^a^	4.41 ^a^	6.73 ^b^	5.94 ^c^	7.28 ^d^	*F*_4,3249_ = 504.91 *; *η*^2^ = 0.20
Safety	6.29 ^a^	6.54 ^b^	5.92 ^c^	5.23 ^d^	7.42 ^e^	*F*_4,3270_ = 147.21 *; *η*^2^ = 0.07

*Note*. Values with different superscript letters (a,b,c,d,e,) in the same row are significantly different from each other at *p* < 0.05. * *p* ≤ 0.001.

## Data Availability

The data presented in this study are available on request from the corresponding author.

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
