# Peer review of "A Cluster Analysis of University Commuters: Attitudes, Personal Norms and Constraints, and Travel Satisfaction"

_ijerph, 2021, doi:10.3390/ijerph18094592_

Round 1

Reviewer 1 Report

This article presents a study on the sociodemographic and psychosocial factors associated with using different modes of transport to respond to sustainable mobility. This study applied a cluster analysis using the SPSS software package. The information is collected through an online survey of a significant sample of the university community (University of Bologna)

It is a well-founded work and supported by references to previous results from other authors. His contributions to state of the art are based on some novel content and concepts studied. From a methodological or technological point of view, the work contributes very little.

Regarding the research procedure, the authors use a standard methodology for this type of study. There is not enough information from the cluster analysis used. A detailed explanation of the procedure, average distance, grouping method is requested, and the reasons why these are used and not others. It would be interesting to accompany the results of the clustering with some graph.

The discussion and conclusions do not show the important discoveries of experimentation regarding strategies for behaviour changes. It also supports them in previous references and gives the impression that they are not the work results.

Author Response

Response to Reviewer 1 Comments

REVIEWER 1

This article presents a study on the sociodemographic and psychosocial factors associated with using different modes of transport to respond to sustainable mobility. This study applied a cluster analysis using the SPSS software package. The information is collected through an online survey of a significant sample of the university community (University of Bologna)

It is a well-founded work and supported by references to previous results from other authors. His contributions to state of the art are based on some novel content and concepts studied. From a methodological or technological point of view, the work contributes very little.

Regarding the research procedure, the authors use a standard methodology for this type of study. There is not enough information from the cluster analysis used. A detailed explanation of the procedure, average distance, grouping method is requested, and the reasons why these are used and not others. It would be interesting to accompany the results of the clustering with some graph.

Response: We have provided further information in Section 2.3 of the rationale behind adopting a two-stage cluster analysis following the methodological guidance of Ballestar et al. (2018), Norusis (2011) and Tkaczynski et al. (2010).  In contrast, we specified the reasons why we adopted specific objective (i.e., trip duration, trip length, number of modes) and behavioural (i.e., mode-use share) variables to cluster the sample of academic commuters. Indeed, based on the findings of Molin et al. [28,30] and their input for future research, the aim of the current study was to identify groups of travellers based on their commuting travel behaviour and their primary mode of transport instead of segmenting travellers based on their demographic, attitudinal, habitual and normative factors as done previously.

Finally, following your suggestion, we added two figures (Figure 1 - 2) to provide an immediate representation of the clustering results.

The discussion and conclusions do not show the important discoveries of experimentation regarding strategies for behaviour changes. It also supports them in previous references and gives the impression that they are not the work results.

Response: Thank you for the suggestion. We agreed with this point and provide more inputs on recommendations and strategies (both organizational as well as political) for behavioural mode shift towards the more sustainable mode of transport. Based on previous findings we have elaborated tailored strategies within each group of travellers.

Reviewer 2 Report

This is an interesting study about commuting trips among the population of a university in Italy. Cluster analysis is used and 5 discrete groups are identifired. Although I found the work interesting, I need more information so us to decide about the recommendation of the paper. Please interpret my decision as major revision since some further work is needed.

Major comments:

  • One major comment is related with the decision to consider the population (staff, administrative, students) as one. The authors should argue if their decision add bias to the outcomes. Population clusters like staff and students at the university are completely different in terms of needs, flexibility, mobility profile etc
  • More information is needed about the existing infrastructure (e.g. parking strategy like preferential parking or not) and the transportation connectivity of university (bus frequency, number of bus lanes connected with the university etc). I am sure that this information affects primarily the final outcomes and should be explicitly mentioned at the limitation part of the study
  • The paper needs more work on the policy implication field. How will someone (dean, university mobility office) utilize the outcomes of the study? Is this work transferable to other universities?  
  • It is very difficult to me, how the content of the paper will fit with the scope of the journal. Please add references that at least include paper submitted at the journal and are close related to the scope and objective of the journal

Minor comments:

  • “because they are friendly to alternative travel modes, have a higher density than other contexts, and offer mixed travel modes.” Please provide some references about this statement
  • Very good sample size. Please explain why you left only 70% of the sample for your analysis. However you should give details about the representatives of the sample
  • I found the literature review weak. Please consider some similar work primarily related with the behavioral stage for change and the attitudes/perceptions of the employees:
    • Politis, I., Papaioannou, P., & Basbas, S. (2012). Integrated choice and latent variable models for evaluating flexible transport mode choice. Research in Transportation Business & Management3, 24-38.
    • Politis, I., Gavanas, N., Pitsiava–Latinopoulou, M., Papaioannou, P., & Basbas, S. (2012). Measuring the level of acceptance for sustainable mobility in universities. Procedia-Social and Behavioral Sciences48, 2768-2777.
    • Papaioannou, P., & Politis, I. (2019). Applying behavior change theory to predict travel behavior of university commuters. In The Practice of Spatial Analysis(pp. 219-251). Springer, Cham

Author Response

Response to Reviewer 2 Comments

REVIEWER 2

This is an interesting study about commuting trips among the population of a university in Italy. Cluster analysis is used and 5 discrete groups are identified. Although I found the work interesting, I need more information so us to decide about the recommendation of the paper. Please interpret my decision as major revision since some further work is needed.

Major comments:

One major comment is related with the decision to consider the population (staff, administrative, students) as one. The authors should argue if their decision add bias to the outcomes. Population clusters like staff and students at the university are completely different in terms of needs, flexibility, mobility profile etc

Response: Thank you for the comment. Indeed, in previous version of the document, our analysis taken into account the different employment status of our sample. We introduced this results in Table 2 and discussed these results in terms of policy implications and organizational actions to promote sustainable behavioural changes (e.g., non-resident students or people who experienced a relocation or a residential change during the last 2 years).

More information is needed about the existing infrastructure (e.g. parking strategy like preferential parking or not) and the transportation connectivity of university (bus frequency, number of bus lanes connected with the university etc). I am sure that this information affects primarily the final outcomes and should be explicitly mentioned at the limitation part of the study

Response: Thank you for this observation. A dedicated paragraph that briefly describes the transportation supply (both in terms of infrastructure and services) connecting the multi-campus university system has been added in the text (section. 2.3). This allows for a clearer interpretation of trip chain and modal preference results.

The paper needs more work on the policy implication field. How will someone (dean, university mobility office) utilize the outcomes of the study? Is this work transferable to other universities?  

Response: Thank you for this observation. In the conclusions, some considerations were added on policy aspects and involvement of university mobility management figures in the development of sustainable mobility plans and projects in urban areas with university campuses, in light of the results obtained in the present research.

It is very difficult to me, how the content of the paper will fit with the scope of the journal. Please add references that at least include paper submitted at the journal and are close related to the scope and objective of the journal

Response: This consideration is very pertinent. The aims and scope of the journal include environmental health, climate change and health, environmental science and engineering. Our work has a strongly interdisciplinary perspective, since it merges methods from social psychology with approaches typical of transport systems engineering, with the aim of investigating the variables that influence the modal choice of university commuters, in order to provide additional elements that can effectively guide mobility policies aimed at increasing sustainability, with clear implications on public health and quality of life. Following the reviewer's instructions, a specific reference was added:

Teuber, M., & Sudeck, G. (2021). Why do students walk or cycle for transportation? Perceived study environment and psychological determinants as predictors of active transportation by university students. International journal of environmental research and public health, 18(4), 1390.

Minor comments:

“because they are friendly to alternative travel modes, have a higher density than other contexts, and offer mixed travel modes.” Please provide some references about this statement

Response: Thanks for the suggestion. We have provided the related references in the text as well as in the reference section.

Very good sample size. Please explain why you left only 70% of the sample for your analysis. However you should give details about the representatives of the sample

Response: As mentioned in the article (Section 2.2), only results from questionnaires fully completed by respondents were included in the analysis. The remarkable size of the sample allowed us to discard questionnaires that were largely incomplete without the need to employ imputation techniques for missing data.

I found the literature review weak. Please consider some similar work primarily related to the behavioral stage for change and the attitudes/perceptions of the employees:

  • Politis, I., Papaioannou, P., & Basbas, S. (2012). Integrated choice and latent variable models for evaluating flexible transport mode choice. Research in Transportation Business & Management3, 24-38.
  • Politis, I., Gavanas, N., Pitsiava–Latinopoulou, M., Papaioannou, P., & Basbas, S. (2012). Measuring the level of acceptance for sustainable mobility in universities. Procedia-Social and Behavioral Sciences48, 2768-2777.
  • Papaioannou, P., & Politis, I. (2019). Applying behavior change theory to predict travel behavior of university commuters. In The Practice of Spatial Analysis(pp. 219-251). Springer, Cham

Response: Thank you for this observation. The literature review has been strengthened by adding references that specifically address commuters' choice of travel strategies in sustainable mobility contexts in academic communities.

Reviewer 3 Report

Thank you for giving me the opportunity to review this paper.

The subject of the paper is interesting, and the research was conducted on a large research sample, which deserves an appreciation.

However, I have a few critical comments regarding the reviewed text, namely:

Research gap and research goal

In the "Introduction: section, the authors quite clearly specify the dichotomous purpose of the article.

The introduction includes a synthetic description of thematically convergent research, but the authors do not conclude this analysis by specifying the identified research gap.

In the further part of the article there are also no clear comments and conclusions regarding this research gap.

Review of the literature and research framework

The structure of the paper does not have the necessary "Literature review" section related to the subject matter: behavioral intentions, attitudes, personal norms and constraints, factors influencing the decision-making process, barriers, behavior models, typology of travelers.

The effect of this lack is the insufficient scope of the literature review. The literature focuses mainly on topics related to transport issues and less to the general theory of consumer behavior.

The lack of this section has another effect - it is not understood on which theoretical framework the authors base their considerations and research assumptions. The text does not present any clear references to any theoretical model of other authors. There is one publication related to Theory of Planned Behavior in the bibliography, but the text does not allow the conclusion that the research assumptions were based on this or maybe some other theory. In the context of the discussed topic, the authors emphasize the possibility of designing sustainable mobility models and the role of universities in sustainable mobility research, but they do not present/verify any research model in this paper.

The lack of research framework and the research model also results in the lack of verbalized research hypotheses.

The structure of the text

The article has an incomplete and disproportionate structure:

- necessary sections are missing: "Literature Review" and  “Research framework”,

- The "Conclusions" section is disproportionately short in relation to the rest of the text and this should be extended,

- In the “Discussion” section, we can find a fragment related to research limitations. This topic, logically related to the proposals for future study directions, usually forms a separate section of the text,

- Due to the specificity of the subject, it seems that the paper should have a "Practical implications" section. Perhaps this section could even include an original concept of an important construct of a sustainable mobility model in the area of ​​profiling travelers and classifying their mobile behavior. The conclusion "Our results increase the theoretical knowledge about commuting modal choice determinants and provide insights for behavioural change programs and organizational policies. In fact, organizational sustainability policies can affect environmental and sustainable attitudes and behaviours, promoting wellbeing and increasing productive behaviours " is quite general and requires more details and presentation of proposals/ideas for using research results in shaping the development of sustainable mobility policy.

Author Response

Response to Reviewer 3 Comments

REVIEWER 3

Research gap and research goal

In the "Introduction: section, the authors quite clearly specify the dichotomous purpose of the article.

The introduction includes a synthetic description of thematically convergent research, but the authors do not conclude this analysis by specifying the identified research gap. In the further part of the article there are also no clear comments and conclusions regarding this research gap.

Response: Thanks for the suggestion. We have specified the research gaps at the end of the introduction section. We added more comments in the “conclusions” section

Review of the literature and research framework

The structure of the paper does not have the necessary "Literature review" section related to the subject matter: behavioral intentions, attitudes, personal norms and constraints, factors influencing the decision-making process, barriers, behavior models, typology of travelers.

The effect of this lack is the insufficient scope of the literature review. The literature focuses mainly on topics related to transport issues and less to the general theory of consumer behavior.

Response:  Thanks for the suggestion. We added a literature review on attitudes, personal norms and constraints, and typology of travelers.

The lack of this section has another effect - it is not understood on which theoretical framework the authors base their considerations and research assumptions. The text does not present any clear references to any theoretical model of other authors. There is one publication related to Theory of Planned Behavior in the bibliography, but the text does not allow the conclusion that the research assumptions were based on this or maybe some other theory. In the context of the discussed topic, the authors emphasize the possibility of designing sustainable mobility models and the role of universities in sustainable mobility research, but they do not present/verify any research model in this paper. The lack of a research framework and the research model also results in the lack of verbalized research hypotheses.

Response: Thank you for this observation. The paper is not based on a specific theoretical model such as TPB. However, we integrated the text with assumptions about research models.

The structure of the text

The article has an incomplete and disproportionate structure:

- necessary sections are missing: "Literature Review" and “Research framework”,

Response: We divided the text into sections based on instructions from the Journal. The introduction included the literature review, a presentation of the research gaps, and our research questions.

- The "Conclusions" section is disproportionately short in relation to the rest of the text and this should be extended,

Response: Thank you. The “Conclusions” section has been extended from 5 lines to 34 lines.

- In the “Discussion” section, we can find a fragment related to research limitations. This topic, logically related to the proposals for future study directions, usually forms a separate section of the text,

Response: We formed a separate section of the text on research limitations.

- Due to the specificity of the subject, it seems that the paper should have a "Practical implications" section. Perhaps this section could even include an original concept of an important construct of a sustainable mobility model in the area of profiling travelers and classifying their mobile behavior. The conclusion "Our results increase the theoretical knowledge about commuting modal choice determinants and provide insights for behavioural change programs and organizational policies. In fact, organizational sustainability policies can affect environmental and sustainable attitudes and behaviours, promoting wellbeing and increasing productive behaviours " is quite general and requires more details and presentation of proposals/ideas for using research results in shaping the development of sustainable mobility policy.

Response: We added a "Practical implications" section. We changed the sentence "Our results increase the theoretical knowledge about commuting modal choice determinants and provide insights for behavioural change programs and organizational policies. In fact, organizational sustainability policies can affect environmental and sustainable attitudes and behaviours, promoting wellbeing and increasing productive behaviours " with more details and presentations of proposals for using research results in shaping the development of sustainable mobility policy.

Round 2

Reviewer 2 Report

I am happy to see that authors managed to treat my concerns in a very adequate way, especially those related with the policy implications and the relativeness with the journal objective

In this respect, I recommend that the paper should be published at the journal

Author Response

Thank you for your insightful inputs and requests. 

Reviewer 3 Report

Thank you for giving me the opportunity to review this paper.

The authors made some changes and additions to the text, but the review of the literature related to the consumer behavior should be expanded (only 4 items were added).

Author Response

Thank you for reviewing our paper. 

We expanded the literature related to consumer behavior (in the text blue color).

thank you for your constructive comments.